# Adherence to Adjuvant Endocrine Therapy in Breast Cancer Patients

**Roberta Rosso, Marta D'Alonzo, Valentina Elisabetta Bounous ***, **Silvia Actis, Isabella Cipullo **, **Elena Salerno and Nicoletta Biglia **

Division of Gynecology and Obstetrics, Department of Surgical Sciences, School of Medicine, University of Turin, 10100 Turin, Italy
* Correspondence: valentinaelisabetta.bounous@unito.it

**Abstract:** Background: Adjuvant endocrine therapy (AET) reduces breast cancer recurrence and mortality of women with hormone-receptor-positive tumors, but poor adherence remains a significant problem. The aim of this study was to analyze AET side effects and their impact on adherence to treatment. Methods: A total of 373 breast cancer patients treated with AET filled out a specific questionnaire during their follow up visits at the Breast Unit of our Centre. Results: Side effects were reported by 81% of patients, 84% of those taking tamoxifen and 80% of those taking aromatase inhibitors (AIs). The most common side effect in the tamoxifen group was hot flashes (55.6%), while in the AI group it was arthralgia (60.6%). The addition of GnRH agonists to both tamoxifen and AI significantly worsened all menopausal symptoms. Overall, 12% of patients definitively discontinued AET due to side effects, 6.4% during the first 5 years and 24% during extended therapy. Patients who had previously received chemotherapy or radiotherapy reported a significantly lower discontinuation rate. Conclusions: AET side effects represent a significant problem in breast cancer survivors leading to irregular assumption and discontinuation of therapy. Adherence to AET may be improved by trustful patient–physician communication and a good-quality care network.

**Keywords:** breast cancer; adjuvant endocrine therapy; adherence to treatment; side effects; tamoxifen; aromatase inhibitors; GnRH agonist; questionnaire





## 1. Introduction

Approximately 80% of breast cancer patients have hormone-receptor-positive tumors. In these patients, adjuvant endocrine therapy (AET) is widely used, which includes tamoxifen or aromatase inhibitors (AIs) with or without GnRH agonists, depending on tumor characteristics and menopausal state. In post-menopausal women, AIs represent the main adjuvant endocrine treatment, as they have demonstrated superior clinical outcomes compared to tamoxifen, while in premenopausal patients, different options are available, such as tamoxifen alone or tamoxifen plus GnRH agonists, with a switch to AIs alone when menopause occurs [1,2]. In particular, in young women with high-risk disease, the addition of GnRH agonists to the aromatase inhibitor exemestane significantly improves DFS and reduces the recurrence rate, as shown by SOFT and TEXT trials [3,4]. International guidelines agree with a standard treatment duration of 5 years, but a 10-year extended therapy may be suggested depending on tumor and patient individual characteristics with the support of specific algorithms such as CTS5 [5]. It has been demonstrated that AET reduces the risk of recurrence by 30% and mortality by 40% in patients with hormone-receptor-positive breast cancer and that extended therapy determines a further reduction, as shown by aTTom and ATLAS trials, as well as MA17R, DATA, IDEAL and NSABP B42 trials [6–14].

Despite these benefits, AET is burdened by considerable side effects and poor adherence to treatment, which represents a significant problem. Regarding side effects, the anti-estrogenic action of tamoxifen causes hot flashes, vaginal dryness, sexual dysfunction and dyspareunia,

while its pro-estrogenic effect on endometrium increases the risk of endometrial hyperplasia, polyps and, rarely, endometrial cancer; moreover, it increases the risk of deep venous thrombosis and pulmonary thromboembolism. On the other side, AIs mostly determine arthralgia, joint pain and osteoporosis, as well as weight gain, headache, insomnia, mood changes and hypercholesterolemia [15,16]. Many clinical trials and epidemiological studies show that side effects have a significant impact on the quality of life and play a primary role in the suboptimal adherence to AET in breast cancer patients [17–20]. The discontinuation rate reported in the literature in the first 5 years of treatment is about 50% with a progressive decrease in adherence from the first year (87%) to the third (79%) and fifth (50%) [3,21–24].

It has been demonstrated that the early discontinuation of AET is related to a decline in survival, increased recurrence risk and reduced DFS, as well as increased medical costs and low quality of life due to disease progression and treatment [25–27].

Another significant element associated with non-adherence to AET is poor patient–physician communication, an inadequate explanation of the type and severity of side effects at the beginning of treatment and poor consideration of them during follow-up visits [28–31]. In fact, many studies highlighted the importance of discussing potential concerns and establishing a trustful patient–physician relationship in the acceptance of AET and adherence to treatment [32–35].

The aim of this study is to analyze the type, incidence and severity of AET side effects and determine their impact on adherence to treatment. We also intend to evaluate the importance of patient–physician communication and the benefit of medical and psychological support strategies.

## 2. Materials and Methods

In this retrospective observational study, we analyzed a population of 373 patients with hormone-receptor-positive breast cancer currently or previously treated with AET (tamoxifen, AI, GnRH agonists). A specific questionnaire was administered to these patients during one of their follow-up visits at the Breast Unit of "Mauriziano Umberto I" Hospital in Turin from January 2021 to December 2021.

The questionnaire was composed of 31 questions and 5 sections: AET tolerance and side effects; adherence to treatment (regularity of assumption, change or suspension of treatment due to intolerance); adherence and tolerance to extended therapy; patient–physician communication and strategies suggested to control side effects; and the importance and efficacy of medical and psychological support (Appendix A).

The study included patients with hormone-receptor-positive breast cancer (luminal A and luminal B) who underwent any type of surgery (mastectomy or conservative surgery) followed by AET (tamoxifen, AI, GnRH agonists) from at least 6 months, also including those in the extended therapy regimen. We did not include in our analysis patients on exclusive endocrine therapy and patients with breast cancer recurrence, nor did we include patients who used both tamoxifen and aromatase inhibitors because it could represent a confounding factor. We did not set any limit in terms of time from diagnosis or from the beginning of follow-up.

## 3. Results

### 3.1. Study Population

Patients' mean age at diagnosis was 59 years, while the mean age at the administration of the questionnaire was 66 years, on average 5.5 years after the beginning of AET. In our sample, premenopausal patients represented 36%, while postmenopausal ones represented 64%. At the time of the administration of the questionnaire, 292 patients (78.3%) had taken AIs, while 81 patients (21.7%) had taken tamoxifen. In total, 73 patients (19.6%) currently or previously used GnRH-agonists—55 of them in association with tamoxifen (75%) and 18 in association with AIs (25%). Seventy-nine patients had extended therapy—90% of them with AIs and 10% with tamoxifen. At the time of investigation, 178 patients (48%) had

been taking AET for less than 5 years, while 195 patients (52%) had concluded the 5-year standard treatment.

Characteristics of patients included in our sample are reported in Table 1.

**Table 1.** Characteristics of the study population.

|  | *n* = 373 |
| --- | --- |
| Mean age (years) | 66.5 (33–90) |
| Mean age at surgery (years) | 59.9 (28–86) |
| Menopausal state at surgery |  |
| Premenopausal | 134 (36%) |
| Postmenopausal | 239 (64%) |
| Type of AET at time of administration of questionnaire |  |
| Tamoxifen | 81 (21.7%) |
| Aromatase inhibitors | 292 (78.3%) |
| Association with GnRH-agonists |  |
| Yes | 73 (19.6%) |
| with tamoxifen | 55 (75%) |
| with AI | 18 (25%) |
| No | 300 (80.4%) |
| Extended therapy |  |
| Yes | 79 (21%) |
| No | 294 (79%) |
| Chemotherapy |  |
| Yes | 158 (42.4%) |
| No | 215 (57.6%) |
| Radiotherapy |  |
| Yes | 256 (68.6%) |
| No | 117 (31.4%) |

### 3.2. Incidence of Side Effects

Eighty-one per cent of patients reported at least one side effect, and the majority of them reported more than one. Side effects were reported by 84% of patients taking tamoxifen and 80% of patients taking AI, and they were described as mild in 43% of cases, moderate in 34% and severe in 23% (Figure 1).

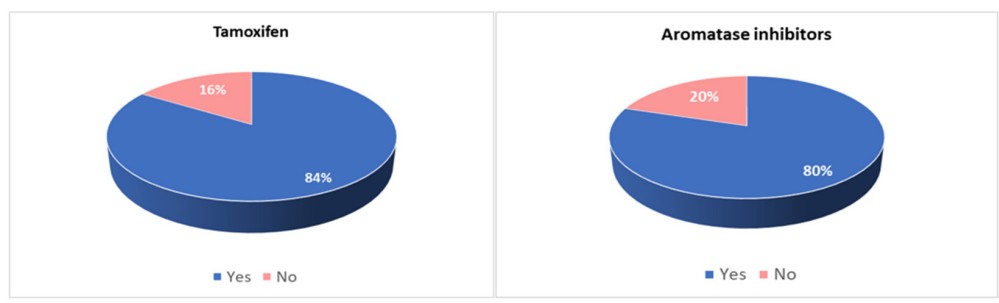

**Figure 1.** Incidence of side effects in patients taking tamoxifen and in patients taking aromatase inhibitors.

Overall, the most common side effects were arthralgia, hot flushes and vaginal dryness. The most common side effects among women taking tamoxifen were hot flushes, arthralgia and vaginal dryness, while they were arthralgia and hypercholesterolemia among women taking AI.

Side effects associated with each therapy and their incidence are reported in Table 2.

**Table 2.** Incidence of side effects of adjuvant endocrine therapy (overall, tamoxifen and aromatase inhibitors).

| Side Effects | Overall | Tamoxifen | Aromatase Inhibitors |
|---|---|---|---|
| | 303 (81%) | *n* = 81 (84%) | *n* = 235 (80%) |
| Arthralgia | 200 (53.6%) | 26 (32.1%) | 177 (60.6%) |
| Hot flushes | 123 (33%) | 45 (55.6%) | 57 (19.5%) |
| Vaginal dryness | 85 (23%) | 21 (25.9%) | 45 (15.4%) |
| Hypercholesterolemia | 70 (18.7%) | 3 (3.7%) | 67 (22.9%) |
| Dyspareunia | 45 (12%) | 9 (11.1%) | 23 (7.9%) |
| Asthenia | 43 (11.5%) | 7 (8.6%) | 41 /14%) |
| Alopecia | 33 (8.8%) | 3 (3.7%) | 28 (9.6%) |
| Weight gain | 16 (4.2%) | 7 (8.6%) | 9 (3.2%) |
| CNS alterations | 14 (3.7%) | 3 (3.7%) | 9 (3.1%) |
| Insomnia | 14 (3.7%) | 3 (3.7%) | 10 (3.4%) |
| Itch | 14 (3.7%) | 3 (3.7%) | 10 (3.4%) |
| Mood changes | 11 (2.9%) | 2 (2.5%) | 8 (2.7%) |
| Liver function abnormalities | 10 (2.9%) | 4 (4.9%) | 7 (2.4%) |
| Headache | 10 (2.9%) | 3 (3.7%) | 5 (1.7%) |
| Decreased libido | 9 (2.4%) | 1 (1.2%) | 5 (1.7%) |
| Dry skin | 6 (1.6%) | 2 (2.5%) | 3 (1%) |
| Thromboembolism | 5 (1.3%) | 4 (4.9%) | 1 (0.3%) |
| Anxiety | 5 (1.3%) | 3 (3.7%) | 2 (0.6%) |
| Dizziness | 4 (1.1%) | 0 (0.0%) | 3 (1%) |

The addition of GnRH agonists to both tamoxifen and AIs significantly increased the incidence of side effects. In particular, patients taking tamoxifen plus GnRH agonists more often reported hot flushes, vaginal dryness, arthralgia and dyspareunia, while patients taking AIs plus GnRH agonists more often reported hot flushes, vaginal dryness, dyspareunia, mood changes, decreased libido and anxiety (Figure 2).

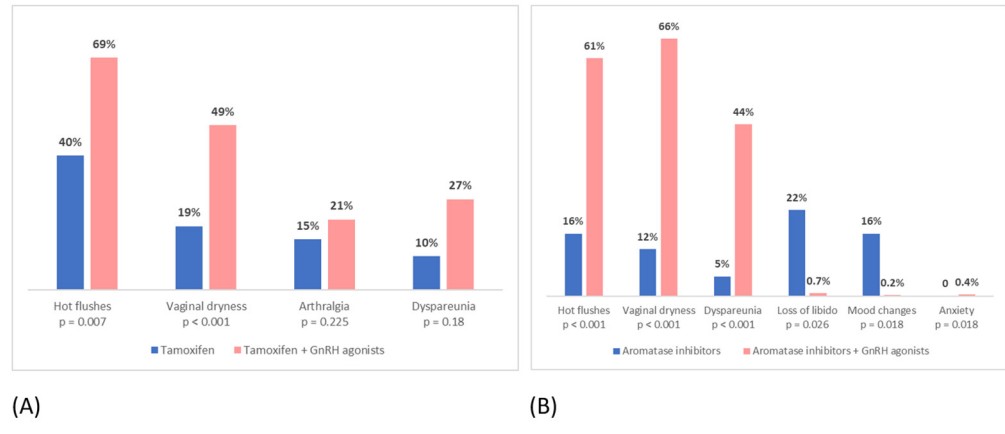

(A)                                                                                          (B)

**Figure 2.** Incidence of side effects in: (**A**) patients taking tamoxifen vs. patients taking tamoxifen + GnRH agonists; (**B**) patients taking aromatase inhibitors alone vs. patients taking aromatase inhibitors + GnRH agonists.

Patients who received adjuvant chemotherapy before starting AET reported a significantly higher incidence of side effects (84.8% vs. 78.6%; *p* < 0.001), while no significant difference emerged between patients who received radiotherapy and those who did not receive it (81.6% vs. 80.3%; *p* = 0.225).

Premenopausal women were more likely to report side effects compared to those who were menopausal at diagnosis (92% vs. 75%; *p* < 0.001). Hot flushes, vaginal dryness, dyspareunia and decreased libido were more frequent and less tolerated by premenopausal

women, while postmenopausal ones reported arthralgia as the most annoying side effect (Figure 3).

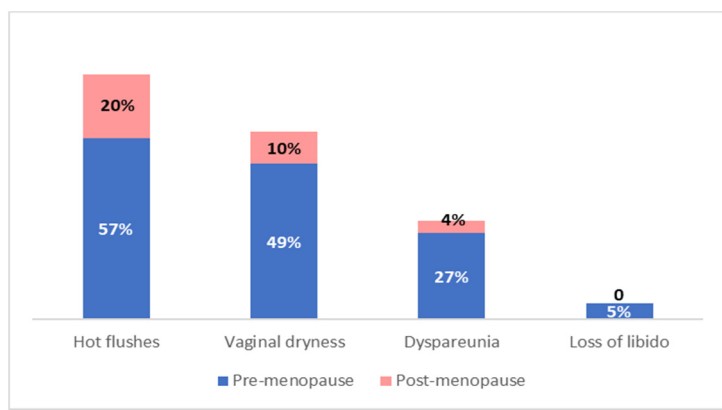

**Figure 3.** Adjuvant endocrine therapy side effects incidence in relation to menopausal status.

In total, 86 of our patients were offered to continue AET for a further 5 years in an extended therapy regimen, and 79 of them (91%) accepted, while 7 refused. Comparing these two groups of women, it emerged that those who refused the extended therapy regimen more often reported moderate and severe side effects during the first 5 years of treatment ($p < 0.001$). Among patients who accepted the extended therapy, 86% had side effects and 16% reported a worsening of them over time.

### 3.3. Adherence to Treatment and Discontinuation

Due to side effects, 79 patients (21%) considered discontinuing AET—57 (72%) taking AI and 22 (28%) taking tamoxifen. In addition, 33 patients (8.2%) reported an irregular assumption—23 (70%) taking AI and 10 (30%) taking tamoxifen. Fifty-nine patients (16%) replaced the treatment with another type of endocrine therapy due to intolerance, while forty-five patients (12%) definitively discontinued the treatment for this reason. The discontinuation rate was 6.4% during the first five years of treatment versus 24% during extended therapy, with no significant difference between different types of AET (14.8% among patients taking tamoxifen and 11.3% among patients taking AIs) (Figure 4).

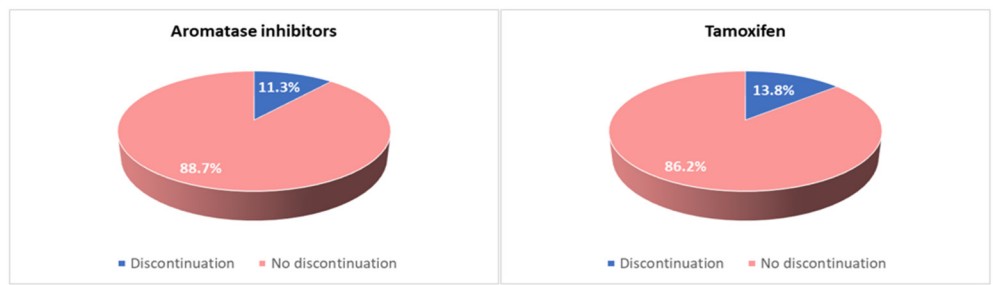

**Figure 4.** Discontinuation rate in patients taking tamoxifen and in patients taking aromatase inhibitors.

Among patients who definitively discontinued the treatment, 24 (53%) did it during the first 5 years, 7 (16%) did not accept extended therapy at the end of the first 5 years of treatment and 14 (31%) discontinued the therapy between the fifth and tenth year, due to intolerance (Figure 5).

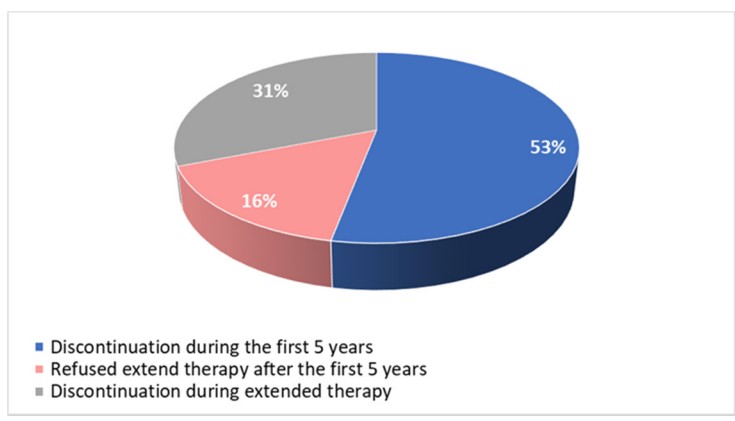

**Figure 5.** Time of discontinuation of adjuvant endocrine therapy.

Overall, 7/45 patients (16%) discontinued AET because of the appearance of severe pathologies, such as endometrial cancer, endometrial thickness or neurologic toxicity, while 38 patients (84%) discontinued because of intolerance to side effects, in particular, arthralgia (64%), hot flushes (4%) and mood alterations (2%), while 11% discontinued treatment for general intolerance, without specific symptoms.

Women who discontinued treatment more often reported severe side effects compared to those who did not discontinue it (44% vs. 15%; $p < 0.001$). Arthralgia was the principal side effect that caused patients to discontinue the treatment (64%).

Patients who had previously received adjuvant chemotherapy showed a lower discontinuation rate, despite a higher incidence of side effects. Even the patients who had received radiotherapy had a lower discontinuation rate compared to those who had not received it (Figure 6).

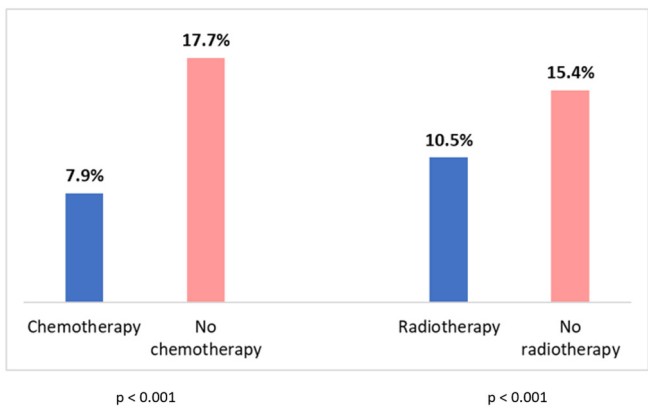

**Figure 6.** Discontinuation rate in patients who have received adjuvant chemotherapy or radiotherapy vs. patients who have not received adjuvant chemotherapy or radiotherapy.

We also stratified our patients by different breast cancer histological subtypes: ductal, lobular and others. The incidence of lobular breast cancer in our population was 10.5%, which is very similar to the incidence in the general population reported in the literature. We did not find any statistically significant difference in the incidence and severity of side effects nor in AET discontinuation rate between the different histological subtypes.

### 3.4. Patient–Physician Communication and Support Strategies

Eighty-eight per cent of patients who experienced AET side effects reported talking about it with the gynecologist during follow up visits. Overall, 77% of patients reported that the gynecologist asked them first about side effects and therapy compliance (Figure 7).

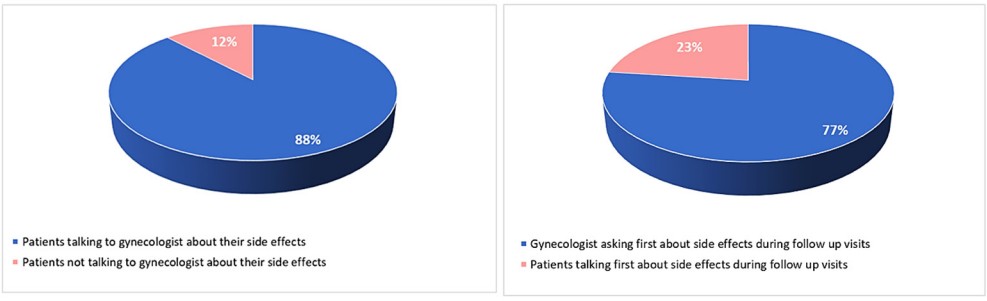

**Figure 7.** Patient–physician communication.

Among women who reported side effects, only 44% took medical treatments to overcome them, especially in the case of vaginal dryness (58.8%), arthralgia (26%) and hot flushes (16%), but 41.3% of them did not report any relief.

Only 9% of patients who experienced menopausal symptoms made regular visits to the dedicated menopause service of the Breast Unit.

Fifteen per cent of patients received psychological support from the dedicated psychology service of the Breast Unit, and ninety-four per cent of them reported that it was very useful and that they would have recommended it to other women diagnosed with breast cancer.

Overall, 94% of women felt well supported during follow-up visits and reported being correctly informed by gynecologists about adverse side effects of AET. On the other hand, 12% of patients who discontinued treatment reported that they would have continued it, if they were better informed about side effects and possible therapies to control them.

## 4. Discussion

The majority of breast cancers are represented by hormone-receptor-positive tumors, and treatment with AET has shown great advantages in terms of disease recurrence and mortality. Despite these widely demonstrated benefits, AET is burdened by considerable side effects, especially in young women. Many clinical trials and epidemiological studies have shown that these side effects significantly impact quality of life and play a primary role in suboptimal adherence to AET [6–13].

Some recent studies have demonstrated that about 90% of patients on AET report side effects, which are more frequent in women taking tamoxifen than in those taking AIs [36–38]. According to these data, side effects were reported by 82% of our patients overall and, in particular, by 84% of those taking tamoxifen and 80% of those taking AIs. Moreover, the literature shows that arthralgia is reported by about 40% of patients taking AIs and 28% of those taking tamoxifen [39]. In our study, arthralgia was the most common side effect, reported by 53% of patients overall and, in particular, by 60% of those taking AIs and 32% of those taking tamoxifen. A possible explanation for this difference is the higher adherence to treatment of our patients, as arthralgia is a symptom that persists and worsens over time during AET. Hot flushes are reported by about 60–70% of patients taking AET in the literature, while in our study, they were reported by only 33% of patients. This difference may also be attributed to the higher adherence to treatment of our patients, as hot flushes tend to decrease in intensity and be more tolerated by women over time [40,41].

In our study, we evaluated the impact of GnRH agonists on the tolerance of tamoxifen and AI. It emerged that the addition of GnRH agonists to both tamoxifen and AI in high-risk premenopausal women significantly increased all analyzed side effects and worsened the tolerance to treatment overall. In particular, the most reported and less tolerated side effects among patients taking tamoxifen or AIs plus GnRH agonists, compared to those taking tamoxifen or AIs alone, were hot flushes, vaginal dryness, dyspareunia, mood changes and decreased libido. Our results are similar to those that emerged from the SOFT156 trial, which demonstrated that the addition of GnRH agonists to both tamoxifen and AIs was

related to a higher incidence of hot flushes, mood changes, vaginal dryness and decreased libido [42].

As there is consistent evidence of poorer AET tolerance in premenopausal women compared to postmenopausal ones, we evaluated side effects and tolerance in relation to the menopausal state [43]. The same results emerged from our study, as premenopausal women reported more side effects than those who were postmenopausal at the time of diagnosis (92% vs. 75%). Younger women more often reported hot flushes, vaginal dryness, decreased libido, dyspareunia and endometrial modifications.

In our study, the discontinuation rate was lower than that reported in the literature (12% vs. about 50%) [21,22,24,28]. It has to be considered that there are different methods of evaluation of adherence to AET in different studies. Self-assessment with a questionnaire or certified scales is the most used method, but in some cases, questionnaires include variables such as the percentage of tablets taken out of the total and consider as "non-adherents" those patients who take therapy irregularly or who simply report side effects [44], while we only considered non-adherent patients who definitively stopped the treatment. In fact, another study in which discontinuation is considered to be definitive suspension of treatment showed a discontinuation rate of 10%, although on a small sample of women [45].

Moreover, it emerged from our study that most of the patients who discontinued AET made this decision when extended therapy was proposed after the first five years of treatment or during extended therapy itself. The refusal of extended therapy may be explained not only by side effects, but also by the fact that patients could have perceived extended therapy as optional, without perceiving its real importance, maybe due to poor communication with the specialist. On the other hand, the discontinuation of AET during the extended therapy, after the initial acceptance, may also be explained by the worsening of the severity of side effects over time, as reported by 16% of our patients.

According to the literature, in our study, arthralgia was not only the most common side effect, but also the main reason for the discontinuation of therapy. A recent metanalysis, in fact, reported that the side effect most related to the discontinuation of AET was arthralgia, followed by weight gain and mood changes, while hot flushes, although very common, were considered physiological and generally did not lead patients to discontinue the treatment [36].

Moreover, from our analysis, it is evident that women who received adjuvant chemotherapy or radiotherapy had a lower discontinuation rate, despite a higher incidence of side effects in women who received chemotherapy. These data are in accordance with a recently published study and may be explained by a higher awareness of the severity of the disease, especially in patients who received chemotherapy, and of the importance of adjuvant treatments by these women [24].

It has been widely demonstrated that patient–physician communication plays a primary role in adherence to any medical treatment. Concerning AET, prior studies report that patients who have a referral specialist (gynecologist or oncologist) showed a higher compliance to AET than those followed up by a general practitioner, probably because the specialist can give the patients more detailed information about the importance of therapy and provide more specific medications to overcome side effects if needed, and this probably helps patients to continue the treatment [36,46,47]. In fact, the Necessity Concerns Framework (NCF) demonstrated that patients' adherence to treatment is related to their perception of the necessity and importance of treatment itself and the reduction of concerns about it more than to its side effects [48]. In fact, a recent study observed a meaningful difference in the necessity beliefs between women who accepted versus those who refused or discontinued AET, showing that women with ongoing AET intake had significantly higher trust in their treating physician and lower concerns regarding AET [35].

Concerning the evaluation of medical and psychological support, it emerged from our study that almost all of the patients taken in at our Breast Unit felt well supported. This may be attributed to a well-organized healthcare network, which allows us to take care of patients globally with regular follow-up visits and a direct communication channel

managed by a breast nurse. Patients have the possibility to refer to them with their problems and organize an appointment with the gynecologist to manage side effects or evaluate the possibility of changing their treatment. All these elements encouraged patients to talk with the gynecologist about their difficulties and side effects and try to manage them together, before discontinuing therapy by themselves. In our Breast Unit, patients are regularly monitored by a gynecologist, who is probably more aware of gynecological and sexual AET side effects and could manage them more easily than another specialist. Moreover, the gynecologist who follows up breast cancer patients in our Breast Unit is often the same person who performed their surgery, and this may result in a stronger patient–physician relationship and continuity of care, which may contribute to the improvement of patients' adherence to treatment.

## 5. Conclusions

Despite its important advantages, AET is burdened by considerable side effects, which represent a significant problem in BC survivors leading to irregular assumption and discontinuation of therapy, especially in young premenopausal women and during extended therapy. Additionally, women who received adjuvant chemotherapy showed a higher incidence of AET side effects compared to those who did not receive it, but at the same time, they have a lower discontinuation rate, as well as women who received radiotherapy.

Moreover, it is evident from our study that adherence to AET may be improved by trustful physician–patient communication and a good-quality care network, which support women during each step of adjuvant therapy.

The challenge is to be more aware of treatment-related side effects reported by patients, to consider therapies to improve their tolerance and to provide patients with dedicated services, offering them adequate medical and psychological support.

The major strengths of our study are the large sample of patients and the fact that they were all taken in at our Breast Unit, thus limiting the differences in their follow-up and management. On the other hand, the main limitation is that our study is a one-time cross-sectional assessment, and our results came from self-reported information, even though complete anonymity was guaranteed to the patients to avoid untrue answers.

**Author Contributions:** All authors have made substantial contributions to the study conception and design. Material preparation, data collection and analysis were performed by E.S., S.A. and R.R. The first draft of the manuscript was written by R.R., I.C. and M.D., and all authors commented on previous versions of the manuscript. V.E.B. performed the revision of the final manuscript, which has been read and approved by all authors. The entire study was conducted under the supervision of N.B. All authors have read and agreed to the published version of the manuscript.

**Funding:** This research received no external funding.

**Institutional Review Board Statement:** Not applicable.

**Informed Consent Statement:** Informed consent was obtained from all subjects involved in the study.

**Data Availability Statement:** The datasets generated and analyzed during the current study are not publicly available due to privacy reasons but are available from the corresponding author on reasonable request.

**Conflicts of Interest:** The authors declare no conflict of interest.

## Appendix A. Questionnaire

1. Did you experience any AET side effect? Yes—no.
2. What side effects did you experienced? Open answer.
3. How would you define your side effects? Mild—moderate—severe.

4. Have you ever considered to discontinue AET because of side effects? Yes—no.
5. Have you ever taken therapy irregularly because of side effects? Yes—no.
6. Have you ever changed your AET because of side effects? Yes- no. If yes, what? Open answer. Did you notice any improvement? Yes—no.
7. Did you stop AET because of side effects? Yes—no. If yes, when? Open answer. Did you stop therapy by yourself or under medical supervision? By myself—under medical supervision.

8. After the first 5 years of AET, was extended therapy suggested to you? Yes—no. If yes, did you accept? Yes—no.
9. Did you experience different or worse side effects during extended therapy? Yes—no. If yes, what? Open answer.
10. Did you stop extended therapy before 10 years of treatment because of side effects? Yes—no. If yes, when did you stop and why? Open answer.
11. If you did not accept extended therapy, it was because of side effects? Yes—no.

12. Have you ever talked to your gynecologist about these problems? Yes—no.
13. Have you ever taken any medication to overcome these symptoms? Yes—no. If yes, what? Open answer. Did you get relief? Yes—no.
14. If you had received more information about side effects by your gynecologist, would you have continued AET? Yes—no.
15. If you had received an effective therapy against your symptoms, would you have continued AET? Yes—no.

16. Have you ever used the menopause service of the Breast Unit? Yes—no. If yes, did you find it useful? Yes—no. Would you recommend it? Yes—no.
17. Have you ever received psychological support? Yes—no. If yes, did you find it useful? Yes—no. Would you recommend it? Yes—no.
18. Did you felt well supported by medical staff during your therapy? Yes—no.

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
