# Peer review of "Adherence to Adjuvant Endocrine Therapy in Breast Cancer Patients"

_curroncol, doi:10.3390/curroncol30020112_

Round 1
Reviewer 1 Report
Your paper investigates a very important area and one that needs more good data. Unfortunately, your paper has many serious issues that limit enthusiasm.
1. Throughout the paper, English wording is not correct in many places, many times that lead to confusion - for example, line 28 - what is meant by take significant advantage?
Did you assess time since diagnosis and time starting tam or AI?
You mention quality of life throughout the manuscript - did you measure quality of life of the women? You cant make these statements without assessing quality of life.
How many times were women assessed - it states during their follow-up visits (line 74) just once or multiple time points?
Did any women use both Tam and AIs?
Did you conduct and multivariable analyses? Are all your analyses univariable? Not very strong without controlling for many variables.
Limitation - this is a one-time cross-sectional assessment - correct? this should be listed as a limitation.
Appendix A - How were the responses to the questions coded?
Author Response
Thank you for the comments.
Below our responses:
Throughout the paper, English wording is not correct in many places, many times that lead to confusion - for example, line 28 - what is meant by take significant advantage?
We rewrote the sentence more clearly in line 28.
Did you assess time since diagnosis and time starting tam or AI?
We did not assessed time since diagnosis and the beginning of AET, but we included only patients who had started AET from at least 6 months, as stated in line 86-88.
You mention quality of life throughout the manuscript - did you measure quality of life of the women? You cant make these statements without assessing quality of life.
In our study, we have not performed any QoL assessment using validated questionnaires, so we eliminated this expression in line 68 and 81.
How many times were women assessed - it states during their follow-up visits (line 74) just once or multiple time points?
We introduced a sentence explaining that the questionnaire was administered once, in line 75.
Did any women use both Tam and AIs?
We excluded women who used both tamoxifen and AIs, because it could be a confounding factor, but we will consider this point for further publications. We introduced a sentence explaining this point in line 86.
Did you conduct and multivariable analyses? Are all your analyses univariable? Not very strong without controlling for many variables.
We are aware that the lack of multivariable analyses reduces the reliability of our observational paper, however we consider this work more of a generator of hypothesis. We look forward to expanding the database and strengthening the statistical work in further publications.
Limitation - this is a one-time cross-sectional assessment - correct? this should be listed as a limitation.
We listed this as a limitation in line 325.
Appendix A - How were the responses to the questions coded?
We introduced the possible answers of any question in Appendix A.
Reviewer 2 Report
Summary
Authors set out to analyze the side effects of adjuvant endocrine therapy on to determine its impact on patient quality of life and their tendency to adhere to the treatment.In addition, they also evaluated the impact and importance of confidence between patient-physician in regards to patients actually adhering to the treatment in a group of 373 hormone receptor positive patients. From the patient responses to the administered questions, authors concluded that addition of GnRH worsened the side effects in both groups of patients (tamoxifen as well as AI). Major strengths of this study are number of patients and consistency in recording the patient feedback. Conclusions made from this study aligned very well with the observations reported by previous trials. In summary, authors believe that lack of adherence to AET can be improved with improved patient-physician communication and good quality care during the treatment regimen.
Overall,
Major points:
1. Can authors try to stratify these hormone receptor positive patients into histological subtypes of breast cancers. Since Invasive lobular carcinoma (about 15% incidence rates) often gets clubbed with luminal BC in spite of being histologically distinct and report their findings, if they align or differ in any sense. This would add additional value to the datasets and these important observations.
Minor points:
1. Spell check on line 46: Dysfunction from disfunction
In summary, this is strong study investigating the impact and importance of patient-physician communication on the adherence to AET regimen in hormone receptor positive breast cancer patients.
Author Response
Thank you for the comments.
Below our responses:
Can authors try to stratify these hormone receptor positive patients into histological subtypes of breast cancers. Since Invasive lobular carcinoma (about 15% incidence rates) often gets clubbed with luminal BC in spite of being histologically distinct and report their findings, if they align or differ in any sense. This would add additional value to the datasets and these important observations.
We conducted this analysis as suggested, but we did not find any significant difference in the incidence and severity of side effects nor in AET discontinuation rate between the different breast cancer histological subtypes. We intorduced this point on line 187.
Spell check on line 46: Dysfunction from disfunction.
We have replaced the word disfunction with the word dysfunction in line 47.
Reviewer 3 Report
This manuscript is generally well structured, and I think it enriches the current research. Nevertheless, it does require some modifications, which I have outlined in the comments below:
Keywords: The missing keyword is “questionnaire”
The inclusion/exclusion criteria for studies should be more precisely presented
Tables 2 and 3 should be merged
All the text: It is not necessary to include the numerical values (% and p values) in the text, which are shown in Figures and Tables, e.g. data shown in Fig. 2 are duplicated in lines 117-124 of the text
The quality of Figures 2 and 7needs to be improved.
Line 210: There is: [36] [37,38]; should be: [36-38]
References should be adapted to the journal's requirements.
Text formatting should be carefully checked.
The language should be modified carefully.
Author Response
Thank you for your comments.
Below our responses:
Keywords: The missing keyword is “questionnaire”.
We have included the word “questionnaire” in the keywords.
The inclusion/exclusion criteria for studies should be more precisely presented.
We have checked and modified the inclusion and exclusion criteria appropriately.
Tables 2 and 3 should be merged.
We have modified and merged table 2 and 3 appropriately.
All the text: It is not necessary to include the numerical values (% and p values) in the text, which are shown in Figures and Tables, e.g. data shown in Fig. 2 are duplicated in lines 117-124 of the text.
We have checked and deleted the duplicated data in the text.
The quality of Figures 2 and 7 needs to be improved.
We have improved the quality of figure 2 and 7.
Line 210: There is: [36] [37,38]; should be: [36-38]
We have modified the citations in line 222 as required.
References should be adapted to the journal’s requirements.
We have adapted references to the journal’s requirements.
Text formatting should be carefully checked.
We have checked the text formatting.
The language should be modified carefully.
We have checked and modified the language appropriately.
Author Response
Thank you for reviewing the manuscript.
Round 2
Reviewer 1 Report
Please check the English throughout the paper.